# A Comparative Study of Field Nematode Communities over a Decade of Cotton Production in Australia

**Oliver Knox [1],\***[ID]**, David Backhouse [1]**[ID] **and Vadakattu Gupta [2]**

[1]    School of Environmental and Rural Sciences, University of New England, Armidale, NSW 2350, Australia; dbackhou@une.edu.au

[2]    Agriculture & Food, CSIRO, Adelaide, SA 5064, Australia; Gupta.Vadakattu@csiro.au

\*    Correspondence: oknox@une.edu.au; Tel.: +61-2-6773-2946

**Abstract:** Soil nematode populations have the potential to indicate ecosystem disturbances. In response to questions about nematode interactions with soilborne diseases and whether genetically modified cotton altered nematode populations, several fields in the Namoi cotton growing area of Australia were sampled between 2005 and 2007. No significant interactions were observed, but nematodes numbers were low and postulated to be due to the use of the nematicide aldicarb. Aldicarb was removed from the system in 2011 and in 2015 funding allowed some fields to be resampled to determine if there had been a change in the nematode numbers following aldicarb removal. No significant changes in the total nematode numbers were observed, implying that the removal of aldicarb had little impact on the total nematode population size. However, an increase in plant parasitic nematodes was observed in both fields, but the species identified and the levels of change were not considered a threat to cotton production nor driven solely by altered pesticide chemistry. Additionally, greater numbers of higher order coloniser-persisters in the 2015 samples suggests that the current cotton production system is less disruptive to the soil ecosystem than that of a decade ago.

**Keywords:** axonchium; helicotylenchus; tylenchorhynchus; pratylenchus; reniform; vertosol; gossypium

## 1. Introduction

The use of pesticides often courts controversy and remains an issue that often results in political intervention [1,2]. Changes in the regulatory processes of both the EU and the United States EPA brought about a decision from Bayer to halt production of aldicarb, a nematicide developed in the 1970s, by 2014 and for complete removal of the product by 2018 [3]. Aldicarb was utilised on a range of crops, but primarily in Australia in cotton, sugar cane and citrus [4,5].

Australian cotton systems have historically been without the nematode related production issues experienced by other cotton producing nations [6], although the presence of the reniform nematode, *Rotylenchus reniformus* [7], in the Theodore production area of Queensland highlights that this status can change. As a consequence of this, aldicarb was not registered for nematode control, but for early season control of aphids, mirids, jassids, mites, wireworms and thrips that aldicarb's systemic activity offered whilst retaining beneficial populations [8]. Control of these early season pests following the removal of aldicarb from Australia in 2011 has been provided either through the optional use of neonicotinoids, in the form of Cruiser® (active ingredient (a.i.) thiomethoxam, Syngenta) [9], or through the continued or adopted use of the organophosphates and carbamates, such as phorate

and carbosulfan, respectively. The impact of neonicotinoids on entomopathogenic nematodes has been reported to have limited impact on reproduction [10,11], which might imply limited effects on other free living soil nematodes [10,11]. The organophosphate and carbamates are known to have nematicidal activity particularly against reniform, lesion and root-knot nematodes [12,13], but existing work has been on sandy soils, not in clay vertosols. Additionally, impacts beyond the targeted pest nematode population have either not been undertaken [12,14] or found no difference [13].

Adoption of the synthetic pyrethroids to control of wireworm and mirids offers protection to above and below ground herbivorous damage, however, their impact on nematodes is negligible [5,15]. This assumption is based on the facts that no deleterious effects from synthetic pyrethroids have been found on entomophathogenic nematodes [16–18]. However, when pyrethroids were introduced to aquatic systems nematodes flourished [19], although *Daptonema trabeculosum* was found to be sensitive to permethrin [15].

In the USA, aldicarb has been replaced in the cotton production system with either Avicta® seed treatments (a.i. abamectin, thiamethoxam, mefenoxam and fludioxanil, Syngenta) in possible conjunction with Velum® (a.i. fluopyram and imadicloprid, Bayer CropScience) or the use of Vydate® (a.i. anticholinesterase, DuPont). At present, these products are not licensed for Australian cotton where rotations and management conditions to promote rapid cotton establishment are the predominant forms of nematode control [20,21].

In our initial nematode work in the Namoi in between 2005 and 2007, the low numbers of recovered nematodes (<5 nematodes/g soil) were hypothesised as being due to the systemic use of aldicarb [22,23]. This assumption was based on the impact aldicarb has on free living nematodes in culture and under carrots [11,24]. However, despite being initially developed as a nematicide, aldicarb has been rarely studied, in relation to free living nematodes [11], does not affect free living nematodes under potato [25] and we could find no published evidence of its impact under cotton rotations. With changes in funding, movement of staff and the removal of aldicarb in 2011, we were unable to test our hypothesis directly, instead resampling fields in in the upper and lower Namoi valley in 2015, which were originally sampled in 2005 and 2007 and for which nematode community analysis had been undertaken [22,26]. The nematode communities were assessed and compared between the sampling years to determine if the nematode numbers had increased with the removal of aldicarb and if there had been changes in the nematode population structure. The results are discussed within the context of the potential for effects on the Australian cotton production system and the ecological significance of the observations.

## 2. Materials and Methods

### 2.1. Soil Characteristics and Nematode Sampling

Field A: In July 2005 and June of 2007, a field in the lower Namoi (field A) was sampled as part of investigations into non-target effects of genetically modified (GM) cotton on soil microbiology. The field soil is a grey vertosol, 52% clay, pH 8.2 and 200 m above sea level. The mean annual maximal temperatures is 26 °C and minimum 12 °C and the area receives 660 mm of summer dominant rainfall. In the field, samples were collected from under each variety being cultivated, resulting in 16 samples in 2005 and 12 in 2007, with sites evenly spaced along 180 m of the plant line. Approximately one kilogram of topsoil was taken to a depth of 15 cm at each site from under mature cotton. In March, 2015, this field was resampled when it was again under cotton, using field maps of the 2007 trial to return to approximately the same location except that only six samples were taken from the plant line at equidistant points from the tail to head ditch with the field having been planted under only one variety. This field had been in a cotton–wheat rotation, with cotton planted in October of every even year. Aldicarb had been applied at cotton sowing at a standard rate of 7 kg Temik®/ha (1.05 kg a.i.) for thrips control with the final application made in October of 2010. In 2012 and 2014, phorate was

applied with cotton sowing as 6 kg Thimet®/ha (600 g/ha a.i.). Neither chemical was used in the wheat phase of the rotation.

Field B: In late October of 2005, soil was sampled from a field in the upper Namoi (field B) as part of an investigation into nematode interactions with verticillium wilt. This field is a black vertosol, 65% clay, pH 8.5 and 270 m above sea level. Mean maximum and minimum temperatures are 12 and 27 °C, respectively, with the area receiving roughly 640 mm of summer dominant rain. One kilogram of surface soil to a depth of 15 cm was recovered from the plant line of cotton seedlings. Briefly, sample points were established from both the Northern and North-Western corners of the field by walking a 20 m by 10 row transect into the crop and taking a sample. The transect walk was then repeated until six samples had been gathered from each entry point. In March, when the field was under mature cotton and again in June of 2015 after picking and root cutting, we collected samples close to the original sampling points, based on field notes and discussions with the farmer. This field had predominantly been under a cotton–cotton–wheat rotation since 1988, although sorghum had been introduced in place of wheat in 2009, 2013 and 2014. Aldicarb had been applied as Temik® at 7 kg/ha in every year that cotton was sown, resulting in aldicarb application in 13 out of 28 years, with the last application in 2011.

Cultivations varied between fields due to differences in the rotations, but both had been subjected to pupae busting, a minimal cultivation to a depth of 10 cm at least 30 cm either side of the plant line, post cotton crop harvesting and had been subjected to bed reformation in the spring prior to cotton planting.

## 2.2. Soil Analysis

In all cases, field sampled soil was placed in plastic bags and returned in a chilled ice box to the laboratory. In the laboratory, the samples were sieved through a 2 mm sieve and a 300 g subsample was sent within 48 h of samples being taken in the field to Biological Crop Protection (Moggill, Queensland, Australia) for nematode community analysis. Briefly, the soil moisture content was determined gravimetrically and 200 mL of soil was weighed and used to establish Whitehead trays for nematode extraction. Nematodes were subsequently recovered from the water solution within the trays and assessed to determine nematode abundance. A sample of approximately 120 nematodes from the count were identified to genus and, in the case of the plant parasitic nematodes, to species where possible to facilitate community compositional analysis [27]. Recovered nematode data were analysed both as recovered numbers and as the number of nematodes present per gram of dry weight equivalent of soil to mitigate moisture and soil porosity differences.

## 2.3. Root Tissue Analysis

Roots were collected from all samples during the sieving process and the root tissue was cleared using the NaOCl and acid fuchsin method of Byrd et al. [26,28]. Roots were spread over a 1 cm gridded Petri dish and examined under a stereo microscope (20 to 45 x magnification) for the presence of nematodes.

## 2.4. Community Comparisons and Statistical Analysis

The nematode community data from the 2005, 2007 and 2015 field samples were tabulated. Comparative analyses for the free living nematodes and between the plant parasitic nematode types were conducted on either raw or percentage compositional data, respectively, with multiple Wilcoxon rank-sum tests between all possible pairwise comparisons. Significance in differences of the median values was taken at the level of $p < 0.05/x$, where x represented the number of groups within any series of pairwise comparisons. This decision was based on the existence of small sample sets for each field and a lack of normality of the data. The nematode channel ratio (NCR) [29] was calculated from the bacterial and fungal trophic group composition of the samples. Additional community composition and change was assessed using the Nematode INdicator Joint Analysis (NINJA) web based

program [30] with probability of similarity of mean outcomes assessed with ANOVA, with significance taken at $p < 0.05$. This on-line tool was also used to generate maturity index (MI), Plant Parasitic Index (PPI), enrichment (EI) and structural indexes (SI) for the samples [31,32].

## 3. Results

### 3.1. Soil Sample and Total Nematode Comparisons

The 200 mL soil samples had an averaged dry weight equivalent of 126.5 g (stdev = 4.5, $n = 30$) for field A and 134.5 g (stdev = 8.8, $n = 18$) for field B over the period of assessment with no apparent statistical difference between weights with sampling time or field, however, moisture content varied between 24% and 35%. The total number of nematodes recovered per 200 mL of soil ranged from 267 to 2944, with an average of 1194, mode of 371 and standard deviation of 609 and standard error of 85. Analysis of the total recovered nematodes did not indicate any significant difference in nematodes/g assessed either within fields, between years or in combination (Table 1), but were detected for many nematode ecological indexes and footprints (Table 2), primarily due to changes in the nematode population structure recorded in 2015 in field B.

**Table 1.** Mean nematode counts of total free living nematodes, per g dry weight equivalent of soil and the percentage of plant parasitic from 200 mL soil Whitehead tray recoveries of samples collected in cotton fields A and B in the Namoi valley. The percentage contributions of the stunt (*Merlinius* and *Tylenchorhynchus* spp.), lesion (*Pratylenchus* sp.) spiral (*Helicotylenchus* sp.) and dagger nematodes to the plant parasitic nematodes within samples and years are given. Similarities in the plant parasitic population are assessed with Wilcoxon rank-sum tests and significantly similar medians are indicated with the same upper case letter.

| Field | Year | Nematodes Per 200 mL | Per g/Soil | % Plant Parasitic | % Stunt | % Lesion | % Spiral | % Dagger |
|---|---|---|---|---|---|---|---|---|
| A | 2005 | 1064 | 8.5 | 2.9 | 92.4 [A] | 1.3 [A] | nd | nd |
|  | 2007 | 791 | 6.3 | 1.0 | 41.8 [B] | 49.9 [B] | nd | nd |
|  | 2015 | 1319 | 10.2 | 1.6 | 11.3 [B] | 81 [B] | 7.7 | nd |
|  |  | ns | ns | ns | $p < 0.01$ | $p < 0.001$ | ns | ns |
| B | 2005 | 1229 | 9.8 | 1.1 | 100.0 | nd | nd | nd |
|  | 2015 | 1687 | 11.8 | 7.7 | 61.7 | nd | 36.8 | 1.4 |
|  |  | ns | ns | $p < 0.001$ | $p < 0.001$ |  | $p < 0.01$* | ns |

\* statistical analysis in cases where the nematode was previously not detected assumes a 0 value in the samples of those years. No detection within the samples is indicated by 'nd' and 'ns' indicates no significant difference.

**Table 2.** Summary mean, standard deviations (SD) and corresponding ANOVA $p$ values from the Nematode INdicator Joint Analysis (NINJA) of the field analysed samples from 2005, 2007 and 2015 in field A and 2005 and 2015 in field B.

| Index Name | | Field A 2005 | Field A 2007 | Field A 2015 | Field B 2005 | Field B 2015 | ANOVA $p$ Value |
|---|---|---|---|---|---|---|---|
| Maturity Index | mean | 2.3 | 2.2 | 2.2 | 2.1 | 2.4 | <0.001 |
|  | SD | 0.2 | 0.2 | 0.1 | 0.1 | 0.1 |  |
| Plant Parasitic Index | mean | 2.4 | 2.6 | 2.1 | 2.3 | 3.2 | <0.001 |
|  | SD | 0.3 | 0.3 | 0.1 | 0.4 | 0.5 |  |
| Enrichment Index | mean | 35.6 | 26.1 | 37.4 | 44.3 | 49.2 | 0.001 |
|  | SD | 6.2 | 17.6 | 14.3 | 10.6 | 12.9 |  |
| Structure Index | mean | 51.3 | 38.5 | 47.5 | 35.5 | 68.4 | <0.001 |
|  | SD | 15.2 | 13.6 | 7.6 | 8.8 | 11.7 |  |
| Nematode channel ratio | mean | 0.7 | 0.9 | 0.9 | 0.7 | 0.9 | <0.001 |
|  | SD | 0.1 | 0.1 | 0.0 | 0.1 | 0.1 |  |

**Table 2.** *Cont.*

| Index Name | | Field A 2005 | Field A 2007 | Field A 2015 | Field B 2005 | Field B 2015 | ANOVA *p* Value |
|---|---|---|---|---|---|---|---|
| Herbivore footprint | mean | 1.5 | 1.5 | 3.2 | 2.0 | 21.0 | <0.001 |
| | SD | 0.6 | 0.6 | 1.4 | 0.8 | 10.5 | |
| Fungivore footprint | mean | 3.2 | 1.3 | 0.7 | 2.2 | 0.8 | <0.001 |
| | SD | 1.5 | 0.7 | 0.5 | 0.8 | 0.5 | |
| Bacterivore footprint | mean | 14.4 | 22.5 | 24.2 | 24.5 | 15.2 | 0.009 |
| | SD | 3.8 | 13.4 | 5.3 | 10.6 | 3.4 | |
| Predator footprint | mean | 1.6 | 0.5 | 0.7 | 0.0 | 0.7 | 0.09 |
| | SD | 1.9 | 1.0 | 0.9 | 0.0 | 0.9 | |
| Omnivore footprint | mean | 11.8 | 7.7 | 5.3 | 6.6 | 5.7 | 0.044 |
| | SD | 8.6 | 3.8 | 2.4 | 5.8 | 2.8 | |
| Total number (nematode/200 mL) | mean | 132.6 | 118.8 | 120.3 | 119.2 | 123.8 | 0.045 |
| | SD | 11.6 | 15.7 | 9.0 | 15.0 | 11.4 | |

### 3.2. Plant Parasitic Nematode Populations

The percentage of the nematode population representing plant parasitic nematodes had not changed in field A and was reflected in the PPI scores for the field, which averaged 2.38, 2.56 and 2.09 for 2005, 2007 and 2015, respectively. However, the PPI had significantly ($p < 0.001$) increased in field B from 2.29 in 2005 to 3.18 in 2015. Additionally, the composition of plant parasitic nematodes, in terms of the abundance of specific parasitic genera, revealed changes in both fields. For example, in the field B there was and remained no evidence of lesion nematodes (*Pratylenchus* sp.), but a significant decrease in stunt (*Merlinius* and *Tylenchorhynchus* spp.) and an increase in spiral (*Helicotylenchus* sp.) nematodes was observed. In field A, spiral nematodes were not observed in 2005 and 2007 samples, but were found in the 2015 samples at >0.2% of the total nematode population. Stunt nematodes were significantly ($p < 0.001$) higher in both fields in 2005 than in other sampling years, whilst the proportion of lesion nematodes increased with time in field A (Table 1). Data on the abundances of the ectoparasites, semi-endoparasites and migratory endoparasites as their % composition of the herbivore assemblage implied that within field A the migratory endoparasites increased as the ectoparasites were reduced, whilst in field B the semi-endoparasties appeared to have replaced the migratory endoparasites (Figure 1).

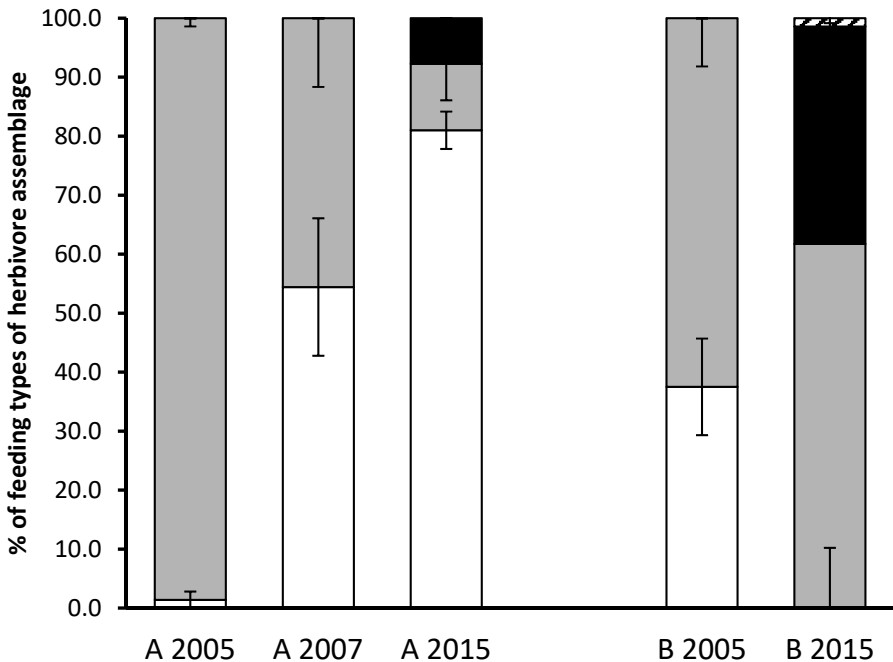

**Figure 1.** The percentage of the migratory endoparasites (e.g., *Pratylenchus*, white), ectoparasitic (e.g., *Tylenchorhynchus*, grey), semi-endoparasitic (e.g., *Helicotylenchus*, black) and ectoparasitic (e.g., *Xiphenema*, dashed) feeding types of the herbivorous nematodes assemblage identified from 200 mL soil samples from field A in 2005, 2007 and 2015 and field B in 2005 and 2015. Number of samples and time of year differed between years with error bars representing the standard error of the means.

### 3.3. Nematode Community Assemblages

Community analysis with NINJA indicated that there was significant ($p < 0.05$, ANOVA) difference in the maturity, plant parasitic, enrichment and structural indexes and the herbivore, fungivore, bacterivore and omnivore footprints within the assessed field material (Table 2). The changes in the assessed community reflected these differences in terms of shifts in the relative proportions of omnivore, predatory, bacterivores, fungivores and herbivorous nematodes present (Figure 2) as well as in changes to the composition of the herbivorous nematode assemblage (Figure 1). Whilst changes in the structural and enrichment status of the samples were both significant (Table 2), graphical representation of the data (Figure 3) supported an improvement in maturity of the analysed ecosystem rather than nutrient enrichment, due to an increase in the number of higher order coloniser-persisters in the samples. This was particularly evident for field B between 2005 and 2015 (Figure 3). NCR analysis indicated similar scores between fields, but that the 2005 samples had a lower ratio than the populations of subsequent samples in both fields (Table 2).

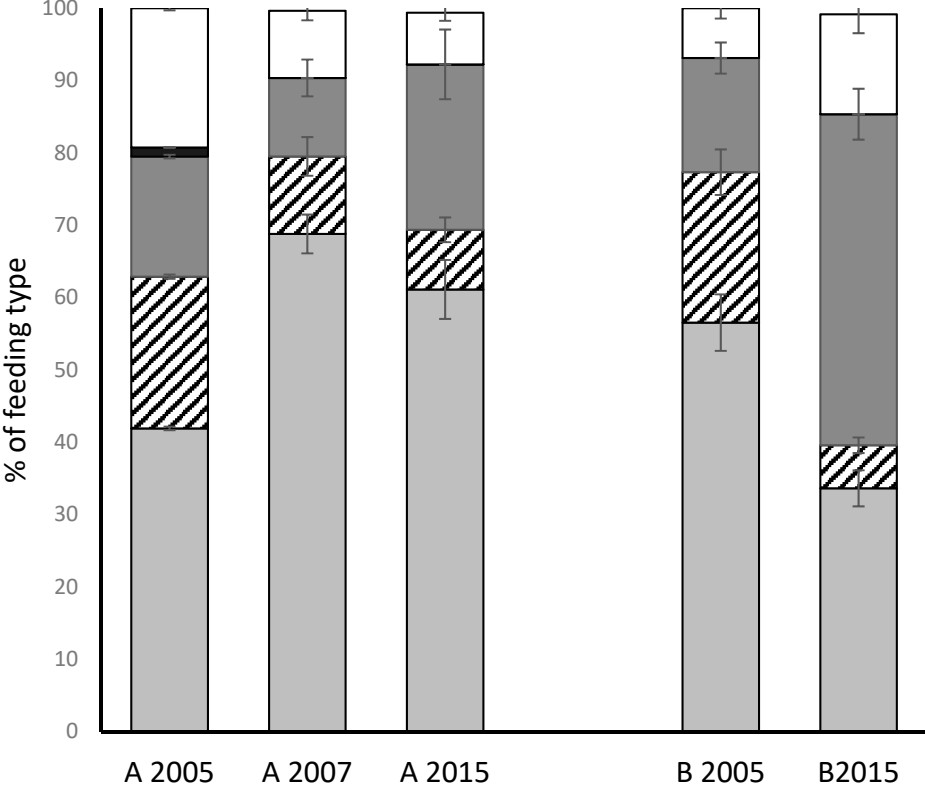

**Figure 2.** Average percentage of the total recovered omnivorous (white), predatory (black), bacterivorous (light grey), fungivorous (stripped) and herbivorous (dark grey) nematode feeding types as identified from the evaluation of ~120 nematodes from each sample (*n* ≥ 6) from field A and field B over each year of sampling. Error bars represent the standard errors of the means.

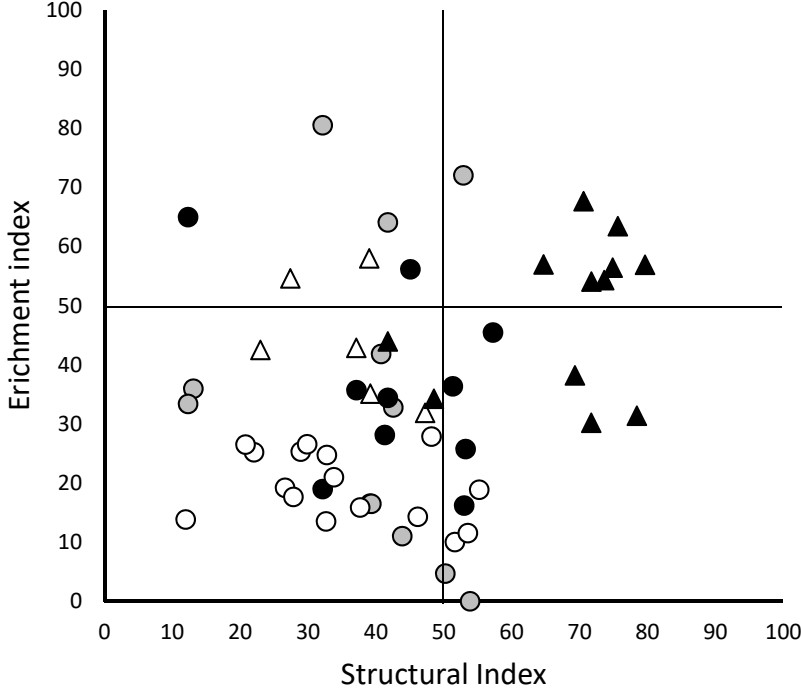

**Figure 3.** Food web analysis of nematode community assemblages from field A, sampled in 2005 (○), 2007 (◓) and 2015 (●) and field B sampled in 2005 (△) and 2015 (▲). The Enrichment index parallels with the nutrient enrichment whilst the structural index correlates with the maturity of the ecosystem.

*3.4. Root Tissue Observations*

No nematodes were observed within the cleared and stained root tissue from the 2015 samples, which was in contrast to the observation and recovery of *H. dihystera* from roots in field B and of Rhabditea and Aphelenchidea from roots in field A in 2005.

## 4. Discussion

In general, abundances of total nematodes in soil supporting Australian cotton systems, as observed in 2005 and 2007 [26,33], are considered low [34]. In addition to this, Australian cotton production systems have not reported nematode issues, with the exception of the recent and localized occurrence of the reniform nematode [7], and this was partly attributed to the widespread use of aldicarb in cotton [22,23]. Aldicarb has a highly variable half-life in soil that ranges from a few to 408 days, with more rapid detoxification occurring in anaerobic soils [4,5]. In Australian cotton soils, the half-life is thought to be about a week in surface soils, due to high soil temperatures and the repeated fluctuation between aerobic and anaerobic soil conditions from flood irrigations [4]. With aldicarb absent from these cotton fields for several years, residual compound and active metabolites from historic applications should have fallen below effective levels [35]. In an attempt to discern if this hypothesis was correct, two fields, roughly 160 km apart, in the Namoi valley, which had nematode community data from 2005 and 2007, were reassessed in 2015.

Although observations from the 2015 sampling indicated that significant changes in the composition of nematode communities were occurring (Figure 3), the total numbers of nematodes supported within the vertosols had not changed (Table 1). This was taken as indication that aldicarb had not imposed a limitation on the population size as initially hypothesised, which is in keeping with other work where pesticide changes had not altered nematode population size, but had been associated with a change in species richness [36,37]. Whilst the implications of other variations in the assessed fields' management systems, such as differences in clay content, irrigation strategies, rotational histories and periods of fallow, could not be investigated from the field records available, it was noted that between the two fields the frequency of fallows occurring post wheat and prior to the return to cotton in the rotation varied [38]. Periods of long fallow of over 7 months in Australian grains production systems, which can incorporate cotton, have been previously reported as causing a reduction in the free living nematode population and altering the nematode channel ratio [29] in favour of a fungal dominated decompositional community [39]. However, the populations analysed in these fields indicated a move to more bacterially dominated decompositional communities over time (Table 2).

Whilst we saw little change in the total free living nematode population across our samples, changes in the nematode community composition were noted in the herbivorous assembly in field B whilst herbivorous nematodes remained unchanged in field A. In a study in Slovakian, maize fields increasing insecticidal chemistry to five times the recommended dose did not significantly alter the nematode communities, but season of assessment did [37]; however, this trial did not interrogate other management decisions. When comparing results from these cotton fields to existing studies [37,39,40], it becomes apparent that there is a requirement for further systematic interrogation of the production systems in order to identify the drivers of nematode community change.

Changes in cotton production practices are also possible causes of the observed differences in the nematode communities within these fields over the last decade [41]. Since 2005, both farms have experienced drought that has seen both differences in the amount and quality of the water used for cotton irrigation in different years, which could have influenced nematode communities [42]. There has also been a change in the preferred cultivar material from cultivars based on the Sicot 189 family in 2005/7 to those of Sicot 74 and 75 in 2015 along with changes in pesticide use and nutrition management [41], which has included the loss of aldicarb from the Australian cotton production system. Additionally, sampling was not possible around the time of aldicarb removal from the system due to funding, staffing movements and that sampling across the two fields occurred at different times

within the cotton phase of the rotation, due to weather constraints that were unavoidable. These issues further highlight that gaps exist in our knowledge of nematodes within Australian vertosols over temporal periods.

Knowledge about the long-term changes in nematode communities due to changing crop management practices would help in the development of options to avoid unexpected threats in addition to providing insights into the ecology of soil fauna in production systems with multiple crop, chemical and physical factors potentially influencing abundance and composition [43]. So whilst the main drivers of nematode community change remain elusive, the nature of the differences between fields and study periods highlighted the continued need for vigilance and the imposition of the 'come clean, go clean' farm hygiene strategy, as currently promoted throughout the Australian cotton industry. This strategy is required to continue to limit the spread of potential problem nematodes, such as the reniform nematode, which is causing cotton production issues in Theodore [7], but remains undetected in New South Wales (NSW). However, the presence of *H. dihystera* within field A and *Xiphenema* sp. in field B in the 2015 samples was noted as neither had been previously detected there. Whilst it is possible that these nematodes were not previously observed due to scarcity, the possibility that they were introduced through soil movement on contaminated machinery over the intervening decade remains plausible.

Changes in other members of the herbivorous nematode population were also noted. *T. ewingi*, was still isolated from both fields, but in field B *T. ewingi* was significantly reduced as a percentage of the plant parasitic population due to an increase in soil recovery of *H. dihystera* (Table 1). This change was hypothesised as being due to rotational differences, which included the incorporation of sorghum into the rotation of field B. This hypothesis was based on both *Tylenchorhynchus* and *Helicotylenchus* spp. being known to survive on wheat [43] and having both been recorded on wheat and sorghum in Australia [44]. Additionally, in a >20 year experiment involving continuous sorghum there was little impact on *Tylenchorhynchus* spp., but incorporation of sorghum straw resulted in a significant increase in the number of *Helicotylenchus* spp. recovered [45], which mirrored the observed change in field B.

The isolation of *H. dihystera* within field B was also noted to have changed over the decade. *H. dihystera* was first observed in Australian cotton roots collected from field B [26], but was absent from the soil samples in 2005. However, these observations were reversed in 2015 with *H. dihystera* only observed in soil. This observation could possibly be linked to the difference in the time of sampling [37] and a reduction in the number of samples, but might also be a function of the maturity of the cotton roots. More likely though is *H. dihystera* ability to feed on sorghum as either an endo or ectoparasite [45,46] and that sorghum was planted into the field B rotation in three of the previous five years to the 2015 sampling.

The other plant parasitic nematode shift considered to be of note was that of the lesion nematode, mostly *P. thornei*, which remained absent in field B, but had significantly increased in numbers in field A. Although still not considered an issue for cotton production in Australia, establishment of a population of around the levels found in 2015 without appropriate management could become an issue for grain crops grown in rotation with cotton [47,48].

Out with the changes in the plant parasitic populations, there was an increase in general maturity index of the community in the 2015 soil samples, suggesting an increase in the abundance of higher order coloniser-persister (C-P) nematodes. This change was particularly evident with the increase in the numbers of *Axonchium* sp., although it was echoed to a lesser extent in other nematodes with C-P scores of >3 [31]. The *Axonchium* nematodes increased from 0.14% to 0.28% of the population in field A, but in field B they increased from 0.42% to 17% of the total population and in some samples represented 40% of the total free living nematode population. Members of the genus *Axonchium* pose something of enigma, because the lack of a clearly identifiable mouth part makes them hard to assign to a specific trophic group. This has seen *Axonchium* associated with either bacterial, root hair and therefore plant parasitic or predatory feeding patterns [49,50]. Given the increase in these nematodes in field B, it would be prudent to establish the exact feeding strategy of these nematodes, as changes in

assignation of feeding strategy to a fungivore or omnivore, rather than an herbivorous ectoparasite, increased the maturity and structural index, whilst reducing the plant parasitic index for field B. However, altering the assigned feeding type for *Axonchium* had little to no impact on either the channel or enrichment index and no effect on field A analysis, where they were less abundant in the samples.

From a production stand point, the apparent rise in plant associated and parasitic nematodes could be seen as grounds for concern, especially in the absence of any chemical or cultivar control options, but at the same time the increase in the maturity index of the populations (Figure 3), partly though changes in predatory nematodes, could be indication of more persistent and stable populations that might self-regulate any potential production threat [37,42]. Although most of the samples still exemplify a state of degradation, based on the quadrat in which they occur [32], there does appear to be a trend toward a trajectory in both enrichment and structural indexes (Figure 3). This observation implies that between 2005 and 2015 the examined cotton production systems are moving toward more opportunistic bacterial feeding strategies, based on the enrichment index, whilst the improvement in the structural index implies a less disturbed soil food web and improved trophic interactions [51]. However, nematodes of the higher order trophic groups, which drive these developments, are known to be easily disrupted by soil cultivation [51], making this a potentially unreliable control mechanism under existing cotton production strategies that still involve some form of tillage.

In general, these observations indicate a continuing change in the nematode populations in the Australian cotton fields sampled, probably due to changes in soil management, rotational variation and seasonal environmental conditions [37,41,45], whereas the impact from pesticides is perhaps not as important as originally hypothesised [22]. However, the scale of the current assessment highlights a need for more intensive sampling and for an improved understanding of the genera present. Whilst changes in the herbivorous nematode populations in these NSW fields implies limited current threat to cotton production in these areas, the risk of movement of the reniform nematode from Queensland and the absence of available nematicidal chemistry would caution that continued monitoring and vigilance is warranted.

**Author Contributions:** O.K. undertook the field sampling, analysis and manuscript preparation. D.B. provided project delivery assistance, technical and editorial support. V.G. assisted with the 2005 and 2007 sample analysis, strategy for the 2015 analysis and manuscript editorial support. All authors have read and agreed to the published version of the manuscript.

**Funding:** This work was undertaken as part of the activities of the Cotton Hub at UNE with funding provided by the University of New England and the Cotton Research and Development Corporation under UNE1403 and UNE2001. The Initial surveys were conducted with funding from the Cotton Catchment and Communities CRC and CRDC with assistance from staff at CSIRO and NSW DPI. Nematode extraction and analysis was conducted by Biological Crop Protection, Moggill, Queensland for all samples other than root material.

**Conflicts of Interest:** The authors declare no conflict of interest.

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
