# Peer review of "A Comparative Study of Field Nematode Communities over a Decade of Cotton Production in Australia"

_agronomy, doi:10.3390/agronomy10010123_

Round 1

Reviewer 1 Report

All my comments and suggestions are included in revised file. I check supplementary files in Excel. There are many question which I have regarding evaulation of data. But try to explain me one, why in list "revised ninja" not al genera are included to analysis?

Author Response

In response to your comment about not all listed nematodes being used in NiNJA, it is because the software processes recognised names in the first instance. Some of the nomenclature we were given in the results did not match and so had to either be reassigned. Hence some of the names differ.

L16: What mean "nematode populations"? Abundance? Trophic structure? Values of ecological indices? Funtional structure? No changes? According wrong ANOVA from Ninja all indices significantly (p=0,001) differed each other, also footprints. Therefore you can not state this

We are referring to the total nematode recovery or nematodes per unit of soil. We have substituded ‘population’ for ‘numbers’ in lines 19 and in an amendment to satisfy reviewer 2. Hopefully this makes the statement clearer.

L20: According to Fig. 2 predatory nematodes were present only in 2005 Field A, and omnivorous abudnance decerased from 2005 to 2015 in field A, on in field B slight increased. Therefore this statement is wrong

It’s not actually wrong as the predatory numbers increased in both fields, although too small to be seen on the graph. The change in omnivores in field A was from 13 to 7 % and not significant, whilst it increased in field B, but given the maturity index is drive by a general change in colonisers and persistent genera we have changed the wording to reflect this.

L60 What mean "low number of nematodes"? Number of total nematodes? PP nematodes? Because in general agricultural soils are charecterised low number of nematodes and diversity of species due to mechanical  disruption of soils, crop rotation, chemical fertilizers etc

Great to hear that someone does not consider our recoveries low. We have added that the original values were <5 nematodes/g soil. When publishing the work in 2006 to 2007 we regularly had US and European reviewers who queried our recovery rates, arguing that they were too low. Have added the number indicated above and a reference to the earlier work.

L65: UNDER COTTON? Crop rotation was with wheat therefore you can not tell under cotton, despite of that you collected soil samples when cotton was grown? You wrote that your study is about effect of aldicarb on free living nemaodes. But majority of results is about PP nematode genus. Maybe there were no PPN species pest of cotton in your fields, therefore there were low number of them, no due to aldicarb appliation? Aldicarb   can be applied against alll PP nematodes? Or against some species only?

OK, so we have added cotton ‘rotations’. You have highlighted the point in your query that we have attempted to make, that whilst aldicarb was developed as a nematicide its actual range of impact on different nematode trophic groups, in different soils and different cropping systems was never fully elucidated. Perhaps, if the chemical had not been removed, this would have been a good topic for a systematic review?

L79: Cotton has been grown also in 2007 here?

Yes and we have amended the text to hopefully reinforce that these samples in 2005 and 7 were taken under a very mature cotton crop. Working for a different CSIRO division at the time our crops were regularly picked late.

L83: Temik was applied to PP nematode control or as insecticide  against thrips, aphids, spider mites, lygus, fleahoppers, or leafminers

For thrips early season control, as stated in the introduction (lines 37-40). Have added ‘for thrips control’ as a reminder. Having never been registered as a nematicide in Australia for cotton the use of aldicarb for nematode control would have be considered off label and ill advised.

L113: All PPN in Results are presented in genus level. Therefore why to species level? Moreover use italics for all genera or species everywhere

Hopefully we’ve now corrected all the italicised issues, many of which occurred during submission and file conversion. Sorry for not picking up on that. The assessment we used, via Biological Crop Protection, always assesses to species level then returns results by common disease issue name and that often covers several genera and species (Table 1), whilst we reported as feeding types as well (Figure 1). In some instances their species report has statements like, pratylenchus probably P. thornei, so having not analysed most of the samples ourselves we would not feel confident reporting to species. We believe that presentation using these terms makes the paper more accessible to a wider audience.

L144: Why total nematodes abundance was expressed on 1 g of dry soil and PPN to 200 ml soil?

We did this to make the data more comparable with our previous studies. Have added the average total counts for the 200 ml recoveries as well. We hope this helps, whilst not giving the mass and GWC of the soils, the difference between the two values is hopefully apparent.

L155 (table 2): This is not good ANOVA from NINJA, becasue this is evaluate means of all parameters together for both fields and sampling date. For example, I don´t believe there was significant differences for p=0,001 among MI 2.3, 2.2 and 2.2 in field A for 2005, 2007, 2015. It is impossible. You can´t take together two fields with different soil type, agricultural management history etc.

The difference is real, whether the data is analysed in NiNJA or off line in another statistical package using the index and footprint values for each sample. It is largely driven by the difference that is observed in the field B samples. We have added, ‘for many of the nematode ecological indexes and footprints analysed (Table 2), primarily due to changes in the nematode population structure recorded in 2015 in field B.’ at the end of the first paragraph of the results (line 153) to highlight this. Hopefully this is not too discursive for the results. The comparison is of the nematode populations and we are not attempting to use the field management or soil metadata in the analysis, which maintains the degrees of freedom to give confidence in the results. To interrogate the data as independent fields would be an alternative option, but you would greatly reduce the confidence in the results. As such we would ask that this table, with these text amendments, be left as is.

L162: italics everywhere

Done

L186: Majority of points depicted in quarat D and C. See Ferris et al. what it mean pls....

We have taken this as a request for a more considered explanation of what is occurring. To address this we have added, ‘Although most of the samples still exemplify a state of degradation, based on the quadrat in which they occur [32], there does appear to be a trend toward a trajectory in both enrichment and structural indexes (Figure 3). This observation implies that between 2005 and 2015 the examined cotton production systems are moving toward more opportunistic bacterial and fungal feeding strategies, based on the enrichment index, whilst the improvement in the structural index implies a less disturbed soil food web and improved trophic interactions [51].’ Into the penultimate paragraph of the discussion.

L197: In Material and methods you wrote that you collected 12 samples in each sampling date. Here you have 16 white circles, 10 black circles, 7 grey circles, 6 white triangels, and 12 black triangels. I seen there big discrepancy, becasue each sympol represent 1 sample or not?

I’ve been back through the data and given access to the old lab files from when the first work was done. There were two fields sampled at the Lower Namoi farm, which caused some confusion, but the reason for the sample differences was to do with changes in the number of cultivars in the trial being investigated between 2005 and 2007, which was not captured in the files I still had access to when first preparing the manuscript. I have amended the methods to reflect this and checked the analysis throughout. Some of the points do overlap. Another way to present the data would be to use the mean and distribution of EI and SI for each year as per the figure in the attached word file.

Is this an improvement in your opinion? Does it show the general trend in MI improvement better and also support the ANOVA in table 2 (basically field B in 2015 was markedly different)? Another alternative would be to undertake a nMDS of the data and plot the 95% confidence intervals of the bootstrap similarity data, but I feel that this moves away from the core data too much.

L199: 80% of all commununities (samples) are placed to qudrat D or close do D which characterised nematode communities or ecosystem maturity as degraded, depleted

Yes, but the general trend is for a move toward more structurally and enriched communities.

Reviewer 2 Report

This study is aimed to sample various nematode populations from the soil of two fields in 3 years. The study has original aspects and provides acceptable amounts of data for possible publication. However, a clearly described objective is missing and some parts of the ms need to be revised before publication.

Comments:

L2: Title is quite general. It should be more specific to the study objective.

L5: Delete 1 before School. Actually you have a double 1.

L10: Abstract: This section gives mainly general statements about the obtained results. Abstract should contain a clear objective and more specific data on the obtained results.

L23: Latin names should be in italic. This should apply throughout the ms.

L28: Please follow correctly journal citation. [1,2]. Follow this throughout the text.

L34: Delete additional space after reniformus

L34: reniformus: r also should be in italic

L35, L38: aldicarb or Aldicarb?

L39: you should distinguished more precisely the name of a.i. and product name. This should apply throughout the text.

L49: A space is missing between ‘negligible’ and ‘(‘.

L56: fluopyram

L71: A clearly stated objective is missing at the end of Introduction

L73: 2.1 is missing before Soil …

L73: nematode

L75: give in full ‘GM’

L105: 2.2 is missing before Soil …

L117: 2.3. is missing before Root … How many roots were collected/examined? Were they replicated?

L121: 2.4. is missing before Community …

L147: Latin names should be in italic. Please prepare this throughout the ms.

L148: These are statistical procedures: should also be in M and M.

L154 2007 and 2015. Give in full SD in the footnote

L159 P is italic or nonitalic. Use one form.

L198: No indication for 2005 and 2015 in the B field.

L203: H. dihystera should be in italic.

L204: Literature should not be cited in Results section (25)

L292: Axonchium- should be in italic.

L331: Reference section is not follow the journal format. This section is negligently prepared.

doi is missing after the cited literatures.

Author Response

Hopefully with the amendments to the introduction and abstract we have met your major concerns with the manuscript. We’ve attempted to address you individual critiques as follows and thank you for your comments and rapid response.

L2: Title is quite general. It should be more specific to the study objective.

We have amended the title and hopefully the change is acceptable while being more specific?

L5: Delete 1 before School. Actually you have a double 1.

Done

L10: Abstract: This section gives mainly general statements about the obtained results. Abstract should contain a clear objective and more specific data on the obtained results.

Have amended the text at line 18 to reflect the purpose by adding, ’allowed some initial fields to be resampled to determine if there had been a change in the nematode numbers following aldicarb removal.’

L23: Latin names should be in italic. This should apply throughout the ms.

Something has gone wrong in the upload, but we have endeavoured to rectify and have found one area where we have definitely dropped the ball on this one. Sorry.

L28: Please follow correctly journal citation. [1,2]. Follow this throughout the text.

We have used the Endnote recommended style for the referencing and the provided manuscript template. In order to attempt to rectify the referencing concerns we have removed the rich text font, re-entered the references and formatted again using the suggested style. This should now all be correct (although we note that DOI is not listed as part of the style guide). We did not track this change on the manuscript.

L34: Delete additional space after reniformus

Done

L34: reniformus: r also should be in italic

Done

L35, L38: aldicarb or Aldicarb?

We have decaptilalised all the active ingredient (chemical) names and capitalised and added the registered trade mark symbol after the trade chemicals, which we have left capitalised. Hopefully this is now clearer?

L39: you should distinguished more precisely the name of a.i. and product name. This should apply throughout the text.

As above.

L49: A space is missing between ‘negligible’ and ‘(‘.

done

L56: fluopyram

done

L71: A clearly stated objective is missing at the end of Introduction

We have added the following at line 81, ‘to determine if the nematode numbers had increased with the removal of aldicarb and if there had been changes in the nematode population structure’

L73: 2.1 is missing before Soil …

Done

L73: nematode

Lower case on nematode? Done.

L75: give in full ‘GM’

Done

L105: 2.2 is missing before Soil …

We’ve added the sub heading numbering as requested.

L117: 2.3. is missing before Root … How many roots were collected/examined? Were they replicated?

We did this for each soil sample in each year. We’ve amended the paragraph to reflect this.

L121: 2.4. is missing before Community …

Done

L147: Latin names should be in italic. Please prepare this throughout the ms.

Done. Hopefully we’ve got them all and will not convert to RTF this time.

L148: These are statistical procedures: should also be in M and M.

It is as the second and third sentences in section 2.4.

L154 2007 and 2015. Give in full SD in the footnote

Given ‘standard deviations’ is mentioned in the legend, is it OK for us to indicate this abbreviation here?

L159 P is italic or nonitalic. Use one form.

Have changed to regular font throughout.

L198: No indication for 2005 and 2015 in the B field.

The symbol loss form the legend has been repaired.

L203: H. dihystera should be in italic.

Done.

L204: Literature should not be cited in Results section (25)

Not a learning I have ever been told before, but we have removed the citation.

L292: Axonchium- should be in italic.

Have italicised Axonchium as the singular genus used throughout, partly because my latin is far too rusty to make plural and present in a nonitlicised form.

L331: Reference section is not follow the journal format. This section is negligently prepared.

I have redone the references throughout the manuscript, partly due to issues with the move to the journal format in the first instance. All entries have been added from Endnote and formatted using the journal’s prescribed EndNote style. We have also checked them since reinserting.

doi is missing after the cited literatures.

This is not in the journal’s recommended style. Are the CrossRef and Pubmed links not added as part of the processing fee?

Round 2

Reviewer 1 Report

Dear authors,

I hope my questions and remarks to your study were negotiable. Neverthelless that you made several changes, I am not satisfy with some explanation, results etc. However, you are responsible for your own works.

Reviewer 2 Report

The authors provided a good revision. The manuscript has been
significantly improved and can be published in Agronomy.